# RECURRENT DEEP DIFFERENTIABLE LOGIC GATE NETWORKS

## ABSTRACT

While differentiable logic gates have shown promise in feedforward networks, their application to sequential modeling remains unexplored. This paper presents the first implementation of Recurrent Deep Differentiable Logic Gate Networks (RD-DLGN), combining Boolean operations with recurrent architectures for sequence-to-sequence learning. Evaluated on WMT'14 English-German translation, RD-DLGN achieves 5.00 BLEU and 30.9% accuracy during training, approaching GRU performance (5.41 BLEU) and graceful degradation (4.39 BLEU) during inference. This work establishes recurrent logic-based neural computation as viable, opening research directions for FPGA acceleration in sequential modeling and other recursive network architectures.

## 1 INTRODUCTION

Large Language Models (LLMs) have become increasingly popular, with applications ranging from everyday productivity tools to scientific research Bommasani et al. (2022). As these models are widely used, the cost of running them has increased. This increase not only leads to higher financial cost, but also raises concerns about energy use and environmental impact Luccioni et al. (2023). Reducing the cost of inference, both in terms of compute and energy, is an important goal for making LLMs more sustainable and widely accessible Patterson et al. (2021).

Deep Differentiable Logic Gate Networks (DDLGNs) are a new way to make neural networks more efficient using the low cost of logic gates. Recent work Petersen et al. (2022; 2024) has shown that DDLGNs can reach strong performance on image classification tasks while using only a small number of active parts. These networks make use of logic-based operations to reduce energy and computation needs. However, current versions of DDLGNs only work

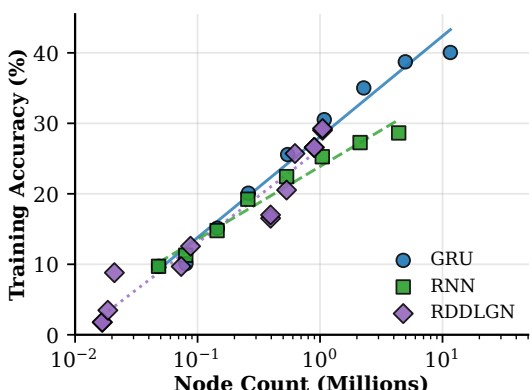

Figure 1: Training accuracy vs. node count (log-scale, 0.01M to 50M) for GRU, RNN, and RD-DLGN models. For RNN/GRU, node count is total parameters minus input embedding; for RDDLGN, it is the sum of all logic layer sizes. Log-linear fit slopes ($R^2$): GRU 14.29 (0.98), RNN 10.34 (0.97), RDDLGN 13.31 (0.94).

with feed-forward and convolutional architectures, and do not use sequential logic elements such as flip-flops or latches. Such elements are common in hardware systems, such as processors and FPGAs, where they help store state and process information over time Patterson & Hennessy (2013).

In this work, we introduce Recurrent Deep Differentiable Logic Gate Networks (RDDLGNs), a new model that brings sequential computation into the DDLGN framework. This idea builds on how recurrent neural networks (RNNs), LSTMs Hochreiter & Schmidhuber (1997), and GRUs Chung et al. (2014) use shared weights over time to model sequences. Newer models, such as Mamba Gu & Dao (2024), also use unrolled temporal computation to capture long-term patterns in language. RDDLGNs follow this principle, combining logic gates with sequential state to support efficient computation over time. We evaluate this new model on the WMT 2014 English-to-German translation

task Bojar et al. (2014) to test translation quality. Our results show that RDDLGNs are a promising step toward developing language models that are more cost-effective and environmentally friendly.

## 2 RELATED WORK

**Deep Differentiable Logic Gate Networks (DDLGN) and Efiicent Architectures** Reducing the computational and energy cost of neural networks is a central concern in academic and industrial settings, given the computational and environmental footprints required to train modern models Luccioni et al. (2023); Faiz et al. (2024). Classic compression methods such as pruning, quantization, and weight clustering Han et al. (2016); Hubara et al. (2018); Gale et al. (2019) have demonstrated that large models can be reduced by orders of magnitude while maintaining accuracy. These techniques serve as foundational tools for model deployment on mobile and embedded devices Liu et al. (2024). For example, extreme quantization can use -1 and +1 for the model weights Courbariaux et al. (2016).

Logic gate networks are a highly energy-efficient alternative to standard neural networks. They replace floating-point matrix operations with sparse binary logic, reducing compute requirements. Unlike traditional networks that rely on many multiply-accumulate operations, logic gate networks use simple two-input gates, such as AND, OR, and XOR. Each gate takes two Boolean inputs and produces one Boolean output, enabling very fast and low-power inference on hardware such as CPUs or FPGAs. Because only a small subset of gates is active per layer, these models achieve high throughput with minimal energy use (Petersen et al., 2022; 2024).

However, training such networks is not straightforward, as the search space of possible logic gate networks is exponentially large, and the gates are discrete and non-differentiable, making them incompatible with gradient-based learning methods, such as backpropagation LeCun et al. (2015). Petersen et al. (2022) overcame these issues by introducing two relaxations (see Section 3 for details). Through this Petersen et al. (2022) match standard Multi Layer Perceptron (MLP) accuracy on MNIST while able to perform over one million inferences per second on a single CPU core.

Later work by Petersen et al. (2024) introduced a convolutional variant of DDLGNs, CDDLGNs, where convolutional kernels were constructed from small trees consisting of logic gates. Thus, they gain the benefits of the conventional operation on spatial data LeCun et al. (1999); Krizhevsky et al. (2012); Ronneberger et al. (2015); Springenberg et al. (2015); He et al. (2016). This allowed the model to achieve 86.3% accuracy on CIFAR-10 while using 29 times fewer logic operations than earlier methods, thus demonstrating that logic-based networks can scale to more challenging tasks while remaining sparse and efficient. Nonetheless, the models still have gradient issues, making model training non-trivial Yousefi et al. (2025).

**Sequential Neural Architectures for Machine Translation** The RNN encoder–decoder framework Cho et al. (2014) uses one recurrent network to read an input sequence into a fixed-size context vector and a second network to generate the output sequence from that vector. While this method improved over phrase-based statistical systems, reaching BLEU scores around 31.20 on English–German benchmarks, it suffered from vanishing gradients and limited ability to model long contexts Cho et al. (2014). To overcome these issues, architectures with gating mechanisms were introduced. Long Short-Term Memory networks (LSTMs) Hochreiter & Schmidhuber (1997) added memory cells and gates to control information flow, improving long-range dependency learning. Gated Recurrent Units (GRUs) Chung et al. (2014) simplified this design by combining gates, achieving similar performance with fewer parameters. Luong et al. (2015) showed in their seminal work that attention can drastically improve language models for machine translation. With the transformer models, recurrence is entirely replaced by self-attention and feedforward blocks, allowing for parallel sequence processing and significantly boosting translation quality Vaswani et al. (2017). Despite these gains, the quadratic cost of self-attention has driven interest in more efficient sequence models. State-space models, such as Mamba Gu & Dao (2024), use linear recurrence and kernel-based parameterizations to capture long-range dependencies with subquadratic complexity. Despite this rich history of temporal architectures, no prior work has explored using discrete logic operations for sequence modeling in neural translation. Our Recurrent DDLGN (RDDLGN) approach fills this gap by embedding sequential logic gates, analogous to flip-flops and latches in hardware, into a differentiable framework. This design aims to inherit the efficiency and interpretability of logic-gate networks while supporting the weight sharing and stateful computation that underlie successful recurrent and state-space models.

## 3 BACKGROUND ON DEEP DIFFERENTIABLE LOGIC NETWORKS

As mentioned earlier, the Differentiable Logic Gate Network (DDLGN) framework did two relaxations to allow gradient-based learning of the networks Petersen et al. (2022).

Inputs are relaxed to real numbers in $[0, 1]$, and smooth surrogate functions replace the Boolean gates. For example, given inputs $x_1, x_2 \in [0, 1]$, AND is relaxed to $x_1 \cdot x_2$, and OR is relaxed to $x_1 + x_2 - x_1 \cdot x_2$. These continuous approximations allow gradients to flow during training.

The choice of gate at each neuron is done through a soft mixture over all 16 possible two-input gates. Learnable logits $z_{j,0\ldots15}$ parameterize a softmax, giving a probability for each gate choice. The neuron output is computed as the weighted sum of all 16 surrogate gate outputs:

$$\phi_j = \sum_{l=0}^{15} \frac{\exp(z_{j,g})}{\sum_{h=0}^{15} \exp(z_{j,h})} \, g_l\big(x_{i_j}, x_{k_j}\big),$$

where $(i_j, k_j)$ index two inputs chosen by a fixed random connectivity and $g_l$ are the 16 relaxed boolean gates.

A `LogicLayer` with width $w$ consists of $w$ neurons in parallel. While the number of trainable parameters for a layer is $16w$ we report the size of a logic layer as $w$, i.e., the effective parameter counts (the number of non-zero weights after training), consistent with prior work on neural sparsity and pruning Gale et al. (2019); Frankle & Carbin (2019); Blalock et al. (2020).

After the final layer, to convert the high-dimensional Boolean outputs into class scores, DDLGN uses a GroupSum operation. GroupSum splits the $w$-dimensional output into $G$ groups (one per class) of size $k$ and computes

$$\text{GroupSum}(\mathbf{x})_g = \frac{1}{\tau} \sum_{i=1}^{k} x_{(g-1)k+i}.$$

This gives a likelihood for each class and softmax can be used to produce normalized predictions.

After training, the inputs are binarized, the gates use the non-relaxed variants, and each neuron is discretized by selecting its highest-probability gate, yielding a fully Boolean network; this network is in "inference mode" and has been `Collapsed`. This enables fast inference and allows the models to be effectively implemented on FPGAs.

For more details, we refer the reader to Petersen et al. (2022; 2024); Miotti et al. (2025).

## 4 METHODOLOGY

### 4.1 MODEL ARCHITECTURE

Our model extends the classic RNN encoder–decoder Cho et al. (2014) by replacing dense layers with differentiable logic gate layers. The encoder ingests a source token sequence and produces a fixed-size context vector. The decoder consumes this context and generates target tokens autoregressively. Figure 2 illustrates the full architecture.

#### 4.1.1 EMBEDDING

Input sequences are first transformed through an embedding layer that maps discrete token indices to continuous vector representations. Given an input sequence of token indices $\mathbf{s} = (s_1, s_2, \ldots, s_T)$ where $s_t \in \{1, 2, \ldots, V\}$ and $V$ is the vocabulary size, the embedding layer produces:

$$\mathbf{E}_t = \text{Embedding}(s_t) \in \mathbb{R}^{d_{\text{emb}}}$$

where $d_{\text{emb}} = 1024$ represents the embedding dimension. Logic gate networks expect Boolean inputs. We therefore relax each token embedding into a continuous value in the range [0,1] by applying a sigmoid:

$$\mathbf{x}_t = \sigma\big(\mathbf{E}_t\big).$$

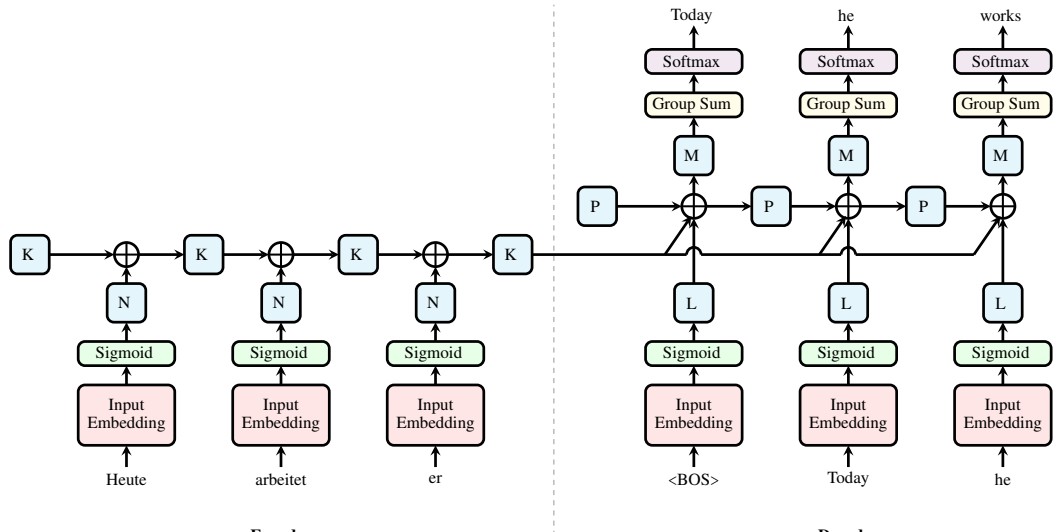

Figure 2: Architecture of the Recurrent Deep Differentiable Logic Gate Networks (RDDLGN). The encoder (left) processes input tokens sequentially through: (1) Embedding layer, (2) Sigmoid activation, and (3) Vertical N-layers. Horizontal K-layers propagate hidden states between $S = 3$ timesteps. The decoder (right) follows a similar structure with L-layers and P-layers, with the shifted target as input tokens. Final encoder state (context vector) is broadcast to all decoder steps. Output probabilities are generated via M-layers, GroupSum followed by Softmax. Color coding: Embeddings (red), Sigmoid (green), GroupSum (yellow), Differentiable Logic Gate Layers (blue), Softmax (violet). Where $< BOS >$ is the beginning of the sentence token.

To encourage the model to produce more decisive binary-like representations, we introduce an embedding regularization loss:

$$\mathcal{L}_{\text{emb}} = \frac{1}{T \cdot d_{\text{emb}}} \sum_{t=1}^{T} \sum_{i=1}^{d_{\text{emb}}} x_{t,i} \cdot (1 - x_{t,i})$$

This regularization term is minimized when embedding values approach either 0 or 1, promoting discrete-like representations while maintaining differentiability. The final embedded representations $\mathbf{x}_t$ are then passed to subsequent layers.

### 4.1.2 ENCODER

The encoder consists of two distinct layer groups: **N-layers** for representation learning and **K-layers** for temporal encoding.

**N-layer Group:** The encoder begins with a group of $D_N$ differentiable logic gate layers that transform embedded input tokens into intermediate representations. Each layer within the N-group applies logic gate operations in feedforward mode:

$$\mathbf{h}^{(n)} = \text{LogicLayer}_n(\mathbf{h}^{(n-1)}), \quad n = 1, \ldots, D_N$$

where $\mathbf{h}^{(0)}$ represents the embedded input tokens, and $\mathbf{h}^{(D_N)}$ is the final N-group output.

**K-layer Group:** Following the N-layers, a group of $D_K$ differentiable logic gate layers processes the sequence temporally from left to right. At each time step $t \in \{1, 2, \ldots, S\}$ where $S = 16$ represents the sequence length, the K-layers maintain a hidden state $\mathbf{k}_t$ computed through the entire K-group:

$$\mathbf{k}_t^{(0)} = [\mathbf{h}_t^{(D_N)}; \mathbf{k}_{t-1}^{(D_K)}]$$
$$\mathbf{k}_t^{(k)} = \text{LogicLayer}_k(\mathbf{k}_t^{(k-1)}), \quad k = 1, \ldots, D_K.$$

$[\cdot;\cdot]$ denotes concatenation, $\mathbf{k}_{t-1}^{(D_K)}$ is the final output of the K group from the previous time step, and $\mathbf{k}_0^{(D_K)}$ is initialized with Gaussian noise. Other strategies can be found in Table 6 in Section C.1.2. The final encoder representation is $\mathbf{c} = \mathbf{k}_S^{(D_K)}$, serving as the context vector.

### 4.1.3 DECODER

The decoder employs three layer-groups: **L-layers**, **P-layers**, and **M-layers** for output generation.

**L-layer Group:** A group of $D_L$ differentiable logic gate layers processes embedded target tokens through feedforward operations:

$$\mathbf{l}_t^{(0)} = \mathbf{y}_t$$
$$\mathbf{l}_t^{(l)} = \text{LogicLayer}_l(\mathbf{l}_t^{(l-1)}), \quad l = 1, \ldots, D_L.$$

$\mathbf{y}_t$ represents the embedded target token at position $t$.

**P-layer Group:** A group of $D_P$ differentiable logic gate layers implements autoregressive decoding through recurrent logic operations. The P-group processes three input sources at each step $t$:

$$\mathbf{p}_t^{(0)} = [\mathbf{p}_{t-1}^{(D_P)}; \mathbf{c}; \mathbf{l}_t^{(D_L)}],$$
$$\mathbf{p}_t^{(p)} = \text{LogicLayer}_p(\mathbf{p}_t^{(p-1)}), \quad p = 1, \ldots, D_P.$$

Where $\mathbf{p}_0^{(D_P)}$ is initialized with Gaussian Noise.

**M-layer Group:** A group of $D_M$ differentiable logic gate layers generates output predictions:

$$\mathbf{m}_t^{(0)} = [\mathbf{p}_t^{(D_P)}; \mathbf{c}; \mathbf{l}_t^{(D_L)}]$$
$$\mathbf{m}_t^{(m)} = \text{LogicLayer}_m(\mathbf{m}_t^{(m-1)}), \quad m = 1, \ldots, D_M.$$

The output of the final M-layer is processed through a GroupSum operation followed by a softmax activation to produce the final probability distribution:

$$\mathbf{r}_t = \text{GroupSum}(\mathbf{m}_t^{(D_M)})$$
$$P(\mathbf{y}_t|\mathbf{x}, \mathbf{y}_{<t}) = \text{softmax}(\mathbf{r}_t).$$

The softmax operation is implicitly computed within the categorical cross-entropy loss during training.

### 4.2 MODEL CONFIGURATION

We determine the final model architecture through a hyperparameter search, with the setup listed in Table 1. Details of the tested configurations and their results are shown in Table 5 in Section C.1. The embeddings for DDLGN need to be higher-dimensional than for the other models, since it works with binary elements instead of continuous values.

Table 1: The number of layers, the layer sizes, and the number of trainable parameters for each layer group. The layer sizes indicate the number of nodes in each `LogicLayer`. The number of trainable parameters (# Parameters) are computed as `sum(sizes) × 16`, and the number of embedding parameters as `embedding_dim × vocab_size`.

| LAYER | LAYER SIZES | # PARAMETERS |
|---|---|---|
| EMBEDDING | 1024 | 16.384M |
| K ($D_K = 2$) | [54K, 32K] | 1.376M |
| L ($D_L = 2$) | [12K, 12K] | 0.384M |
| M ($D_M = 3$) | [400K, 400K, 480K] | 20.480M |
| N ($D_N = 2$) | [12K, 12K] | 0.384M |
| P ($D_P = 2$) | [64K, 48K] | 1.792M |
| **TOTAL** | - | **40.8M** |

### 4.3 LOSS FUNCTION

Our training framework employs a composite loss function combining the primary supervised objective with scheduled auxiliary losses. These auxiliary losses are designed to reduce the discretisation gap by encouraging the model to select a specific gate, helping the DDLGN better approximate the final discrete logic gate network Yousefi et al. (2025).

**Primary Loss Function**  We use categorical cross-entropy with label smoothing Szegedy et al. (2016); Müller et al. (2020) with $\alpha = 0.1$ for the sequence-to-sequence learning task:

$$\mathcal{L}_{\text{main}} = -\sum_{i=1}^{N} \sum_{j=1}^{C} \tilde{y}_{i,j} \log(\hat{y}_{i,j})$$

where $\tilde{y}_{i,j}$ is the label-smoothed target distribution.

**Auxiliary Loss Scheduling**  Our framework supports multiple auxiliary losses with independent scheduling. The primary auxiliary loss employed is binary regularization, which encourages embeddings to approach binary values. This loss is scheduled using a linear ramp from weight $w = 0.0$ to $w = 0.1$ between training steps 1K and 100K:

$$\mathcal{L}_{\text{total}}(t) = \mathcal{L}_{\text{main}} + \sum_{i=1}^{K} w_i(t) \cdot \mathcal{L}_{\text{aux}_i}$$

where $w_i(t)$ represents the time-dependent weight for auxiliary loss $i$. The linear ramp allows the model to first learn basic mappings before enforcing additional constraints on the representations.

## 5 EXPERIMENTS

### 5.1 EXPERIMENTAL SETUP

#### 5.1.1 TRAINING DATA

We train our model on the WMT'14 English-German dataset with 4.5 million sentence pairs Patterson et al. (2021). The data is tokenized at the word level using regex-based splitting, with a shared 16,000-token vocabulary for English and German. This setup was chosen based on the experiments in Table 2 in Section 5.2.

All sequences are truncated or padded to 16 tokens, with longer sequences filtered during preprocessing. Training uses token-based batching with approximately 1024 tokens per batch, dynamically adjusting batch sizes. The preprocessing pipeline filters invalid samples including empty sequences and all-padding samples to ensure training stability.

#### 5.1.2 BASELINES

We evaluate our RDDLGN on the WMT'14 English-German translation benchmark against three baseline architectures (see below). All models were trained under consistent conditions with some architecture-specific optimizations. Table 3 shows that RDDLGN uses orders of magnitude fewer logical operations, allowing lower energy usage.

- **Transformer**: Standard architecture with 2 layers, 8 attention heads, 256-dimensional embeddings, and 1024-dimensional feed-forward layers (16.0M parameters)
- **GRU**: Gated Recurrent Unit encoder-decoder with 256-dimensional embeddings and hidden states (9.0M parameters)
- **RNN**: Vanilla recurrent encoder-decoder with 256-dimensional embeddings and hidden states (8.5M parameters)
- **RDDLGN**: Our differentiable logic gate architecture with 1024-dimensional embeddings and variable layer sizes (40.8M trainable parameters, 1.526M gates, and 16.384M parameters in the encoder, model size 17.91M)

#### 5.1.3 TRAINING CONFIGURATION

All models used a sequence length of 16 tokens, a batch size of 1024 tokens, label smoothing ($\alpha = 0.1$), and AdamW optimization. The Transformer employed learning rate warmup scheduling, while the others used plateau-based learning rate reduction. Baseline models were configured to provide fair comparisons against RDDLGN (more details in Section B)

Table 2: Impact of tokenizer parameters on RDDLGN performance. We report the mean ± standard deviation on the validation set across different sequence lengths (A1-A4), vocabulary sizes and sharing strategies (B1-B7), and tokenization levels (C1).

|  | SEQ LEN | VOCAB SIZE | SHARED VOCAB | TK LEVEL | ACC (%) ↑ | BLEU ↑ | PPL ↓ |
|---|---|---|---|---|---|---|---|
| BASE | 16 | 16000 | TRUE | WORD | 23.28±1.96 | 3.59±0.33 | 209±10 |
| A1 | 4 | | | | 17.17±9.74 | 3.87±1.91 | 3616±959 |
| A2 | 8 | | | | 27.23±3.46 | 5.11±0.93 | 224±28 |
| A3 | 32 | | | | 23.61±0.98 | 3.18±0.20 | 158±5 |
| A4 | 64 | | | | 7.56±13.09 | 0.78±1.28 | 11337±9693 |
| B1 | | 8000 | FALSE | | 25.29±1.22 | 3.94±0.31 | 166±8 |
| B2 | | 8000 | TRUE | | 30.01±0.83 | 4.80±0.05 | 95±3 |
| B3 | | 16000 | FALSE | | 20.42±0.35 | 3.14±0.07 | 290±5 |
| B4 | | 20000 | FALSE | | 17.98±1.73 | 2.68±0.24 | 388±21 |
| B5 | | 20000 | TRUE | | 22.34±1.41 | 3.39±0.25 | 259±14 |
| B6 | | 32000 | FALSE | | 16.53±1.09 | 2.41±0.16 | 540±17 |
| B7 | | 32000 | TRUE | | 18.98±0.48 | 2.82±0.07 | 400±5 |
| C1 | | | | SUBWORD | 11.08±0.25 | 1.76±0.04 | 1241±21 |

## 5.2 DATASET AND TOKENIZER

**Sequence Length (Table 2 Rows A):**  Sequence length affects both context availability and prediction complexity, with accuracy calculated only on non-padding tokens. Very short sequences (A1: 4 tokens) lack context (17.17% accuracy), while very long sequences (A4: 64 tokens) increase prediction errors (7.56%). Moderate lengths (A2: 8 tokens) perform best (27.23%).

**Vocabulary Size and Sharing (Table 2 Rows B):**  Smaller vocabularies reduce classification complexity. Shared vocabularies (B2: 8K shared, 30.01%) consistently outperform separate vocabularies (B1: 8K separate, 25.29%) through better parameter sharing. Performance degrades with larger vocabularies due to increased prediction difficulty.

**Tokenization Level (Table 2 Rows C):**  Subword tokenization (C1: 11.08%) severely underperforms word-level baseline (23.28%). This occurs because subword tokens require more predictions per semantic unit and provide less coherent targets for discrete logic operations.

**Key findings:** Accuracy is sensitive to prediction complexity - shorter sequences and smaller vocabularies reduce error opportunities. Shared vocabularies enable better cross-lingual learning. Word-level tokenization suits logic-based architectures better than subword fragmentation.

## 5.3 LANGUAGE TRANSLATION

Table 3 and Figure 3 presents our experimental results comparing RDDLGN performance in both standard and collapsed configurations against baseline models.

The uncollapsed RDDLGN achieves 5.00 BLEU and 30.9% accuracy, successfully outperforming the RNN baseline (4.59 BLEU) and positioning itself between RNN and GRU performance levels. This demonstrates that differentiable logic gates can effectively model sequential dependencies in translation tasks.

When using the inference model, performance drops to 4.39 BLEU and 27.7% accuracy—a controlled degradation that still achieves good results. The collapse mechanism switches from continuous softmax to discrete argmax operations and uses the Heaviside function instead of sigmoid for embeddings. While the model size technically is larger than any of the baseline models one must remember that out of the 17.91M parameters, 16.384M are used for the embeddings (4 times more than for the baselines). This was done as the embeddings for RDDLGN are binary vectors rather than vectors of floats, thus decreasing their expressiveness. To highlight this difference we show in Table 3 the model sizes with and without the embedding layer for each model.

Table 3: Performance comparison on WMT'14 English-German translation task. RDDLGN (first row) represents the uncollapsed model with standard softmax operations and full embeddings. The collapsed variant uses argmax operations and fully collapsed embeddings, demonstrating controllable performance-efficiency trade-offs. Model sizes are shown with and without embedding parameters; for RDDLGN, we set "Parameters" as the number of trainable parameters, while `Collapsed` shows the model size as discussed in Section 4.2. The collapsed RDDLGN uses far fewer parameters outside embeddings while achieving performance comparable to recurrent baseline models. The table also shows floating-point operations (FLOPs) and logic operations (Logic OPs) required for trained models. For the baseline models, the logic operations are estimated as in Petersen et al. (2022).

| | | | MODEL SIZE | | | |
| MODEL | BLEU ↑ | ACCURACY ↑ | WITH EMB. | WITHOUT EMB. | FLOPs | LOGIC OPs |
|---|---|---|---|---|---|---|
| TRANSFORMER | 5.98 | 35.3% | 16.0 M | 11.9 M | 80.04M | (80.04G) |
| GRU | 5.41 | 34.2% | 9.0 M | 4.9 M | 37.49M | (37.49G) |
| RNN | 4.59 | 29.6% | 8.5 M | 4.4 M | 35.36M | (35.36G) |
| RDDLGN (OURS) | 5.00 | 30.9% | 40.8 M | 24.42 M | – | – |
| COLLAPSED | 4.39 | 27.7% | 17.91 M | 1.53 M | – | 1.53M |

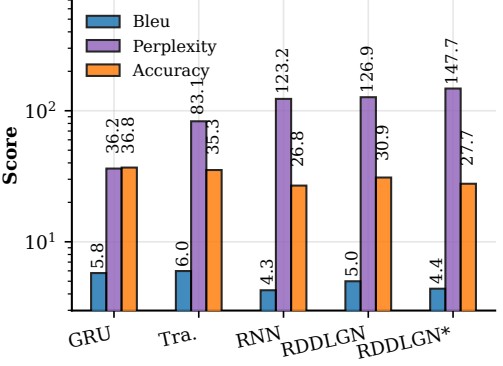

Figure 3: Comparison of test BLEU score, perplexity, and accuracy across the four architectures: **Tra.** (Transformer), **GRU**, **RNN**, and **RDDLGN**. "RDDLGN*" indicates RDDLGN evaluated in the `Collapsed` mode.

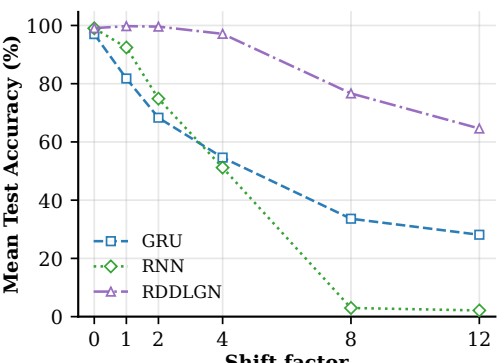

Figure 4: Test accuracy versus shift factor for two of the baseline neural network architectures and RDDLGN. The shift factor controls the tokenization offset applied to input sequences during preprocessing. A factor of 4 means the models need to remember and propagate information over 4 time steps.

## 5.4 MEMORIZATION CAPABILITIES

To further assess the ability of each model to retain and recall information over long sequences, we evaluated the RNN, GRU, and RDDLGN decoders on a shifted monolingual prediction task using a data subset of 0.5M sentences. The shift factor determines how many positions the target tokens are offset with respect to the input, thus probing each architecture's memory span and robustness to temporal distance. For example, with input `[the, cat, sat, on, the, mat]` and shift 2, the target is `[<PAD>, <PAD>, the, cat, sat, on]`.

Our results show that the RDDLGN decoder exhibits substantially enhanced memorization compared to classical recurrent architectures. While RNN and GRU models experience sharp declines in accuracy as the shift factor increases, RDDLGN maintains high test accuracy even for large temporal shifts. For shift factors up to 4, RDDLGN achieves over 97% accuracy, whereas the RNN and GRU baselines drop below 55%. Even at a shift of 12, RDDLGN still records 64.6% accuracy, markedly outperforming the RNN (2.1%) and GRU (28.1%) counterparts, further illustrated in Figure 4.

## 5.5 Gradient Analysis

We assess gradient propagation and training stability in our RDDLGN by evaluating final layerwise gradient statistics. As shown in Table 4, all layer groups (K, L, M, N, P) maintain nonzero and consistently scaled gradients at the end of training, confirming the absence of vanishing or exploding gradients–major pitfalls in deep sequence models. Standard RNNs often suffer from vanishing gradients in the lower layers, but RDDLGN exhibits robust and uniform gradient flow throughout all layers, supporting stable and efficient learning.

Table 4: Final per-group gradient statistics: mean, std, and std/mean for $|\nabla_w \mathcal{L}|$ at training end.

| Layer | Mean | Std | Std/Mean |
|---|---|---|---|
| K0 | 5.90E+04 | 4.68E+05 | 7.936E+00 |
| L0 | 1.02E+04 | 8.11E+04 | 7.931E+00 |
| M0 | 7.37E+04 | 5.85E+05 | 7.932E+00 |
| M1 | 4.83E+04 | 3.83E+05 | 7.936E+00 |
| M2 | 4.83E+04 | 3.83E+05 | 7.937E+00 |
| M3 | 4.83E+04 | 3.81E+05 | 7.884E+00 |
| M4 | 4.83E+04 | 3.83E+05 | 7.933E+00 |
| M5 | 1.20E+05 | 9.52E+05 | 7.936E+00 |
| M6 | 1.20E+05 | 9.52E+05 | 7.935E+00 |
| N0 | 8.82E+03 | 6.90E+04 | 7.819E+00 |
| P0 | 7.82E+04 | 6.20E+05 | 7.927E+00 |

**Gradient Flow Explanation and Expectation**   The gradient flow mechanism in RDDLGN stems from the sum of weighted operators formulation. The gradient with respect to the $i$-th logit $w_i$ is:

$$\frac{\partial y}{\partial w_i} = p_i \left( f_i(\mathbf{x}_1, \mathbf{x}_2) - \sum_{j=1}^{M} p_j f_j(\mathbf{x}_1, \mathbf{x}_2) \right)$$

The expected gradient behavior can be analyzed through its components:

$$\mathbb{E}\left[ \frac{\partial y}{\partial w_i} \right] = \mathbb{E}[p_i f_i] - \mathbb{E}\left[ p_i \sum_{j=1}^{M} p_j f_j \right]$$

This expectation remains non-zero as long as the experts produce diverse outputs ($f_i \neq f_j$ for some $i, j$) and the selection probabilities are correlated with expert performance. The gradient magnitude is proportional to the discrepancy between expert outputs, preventing vanishing gradients when the experts are diverse. Thus, it maintains stable gradient flow throughout all layers during optimization.

## 6 Conclusion

This work presents the first application of differentiable logic gate networks to sequence-to-sequence learning tasks, introducing a novel architectural paradigm for neural machine translation. Our RDDLGN demonstrates the feasibility of replacing traditional neural building blocks with logic-based computation, achieving performance comparable to GRU baselines on WMT'14 English-German translation (5.00 vs. 5.41 BLEU). This competitive performance indicates that logic gate architectures can potentially solve other sequential modeling problems, opening new research directions in neural computation on FPGAs.

The proposed approach faces some challenges. Training requires more parameters for embeddings (16.384M vs. 4.096M for baselines), making the total trainable parameter counts much higher for RDDLGNs. Our recurrent model faces the same issues as the original DLGN, with longer training times to achieve comparable performance to conventional models. Additionally, the architecture suffers from vanishing gradient problems, particularly for longer sequences and deeper layer configurations.

Several promising research directions emerge from this work. Two key technical improvements warrant investigation: weight reparametrization approaches to address vanishing gradients, and hardware implementation through FPGA synthesis to exploit the model's discrete logic operations. Most significantly, incorporating associative recurrent blocks could transform training efficiency from linear to logarithmic complexity ($O(\log n)$ training time, $O(n \log n)$ computational complexity), as demonstrated by Orvieto et al. (2023), potentially making logic-based architectures competitive for long-sequence modeling tasks.

## REPRODUCIBILITY STATEMENT

All code used in our experiments is included in the supplementary material, along with a README describing how to run the training and evaluation scripts. The training and test data are publicly available through PyTorch's torchtext, Kaggle, and Huggingface. The code will be made publicly available on GitHub with the camera-ready version. Details of model architectures, training procedures, and datasets are provided in Sections B and 4.

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

## A    USAGE OF LLMS

We have made use of several large language models (LLMs) during the preparation of this work. ChatGPT, Claude, Gemini, and Grammarly were employed to assist with spellchecking, improving wording, and shortening text for clarity and readability. In addition, ChatGPT, Claude, and Cursor were used for analyzing and explaining code, providing code completions, and generating visualizations to support our implementation and experiments. These tools were applied as auxiliary aids to polish the writing and streamline the development process, while the core research contributions, experimental design, and interpretation of results remain entirely our own.

## B    TRAINING DETAILS

### B.1    OPTIMIZER

Our training framework employs the AdamW Loshchilov & Hutter (2019); Kingma & Ba (2017) optimizer with standard hyperparameters suitable for sequence-to-sequence learning Wu & Xing (2024). Specifically, we use AdamW with a learning rate of 0.05 (managed by a scheduler), weight decay $\lambda = 0.001$ for L2 regularization, momentum parameters $\beta_1 = 0.9$ and $\beta_2 = 0.999$, and the default epsilon value $\epsilon = 10^{-8}$.

The relatively high base learning rate is modulated by the scheduler to ensure stable training.

## B.2 SCHEDULER

Our learning rate scheduling employs a single adaptive mechanism that monitors training progress and adjusts the learning rate globally across all parameters.

**Adaptive Learning Rate Scheduling** We employ a ReduceLROnPlateau Paszke et al. (2019) strategy that monitors validation loss and adaptively reduces the learning rate when training plateaus:

$$\eta(t+1) = \begin{cases} \eta(t) \cdot \gamma & \text{if validation loss plateaus for } p \text{ steps} \\ \eta(t) & \text{otherwise} \end{cases} \quad (1)$$

with reduction factor $\gamma = 0.8$ and patience $p = 10K$ steps. The learning rate is applied uniformly across all parameter groups, with the base rate initialized to $\eta_0 = 0.05$.

# C HYPERPARAMETER STUDIES

This section presents a comprehensive empirical analysis of hyperparameter effects on our RDDLGN model performance. We systematically investigate model parameters, training configurations, stochastic components and dataset-tokenizer interactions.

Each study employs controlled experimentation with multiple random seeds (typically 3) to ensure statistical reliability. Performance is evaluated using accuracy (ACC), BLEU score, and perplexity (PPL) on the WMT14 German-English translation task with training for only for 1 epoch on $10\%$ of the dataset. Results are reported as mean ± standard deviation to capture both central tendency and variability.

## C.1 MODEL PARAMETER

We systematically investigate architectural configurations using our baseline model.

### C.1.1 LAYER SIZES

**N-layers (Table 5 Rows A):** Representation learning layers show modest variations around baseline performance. Configuration A4 [20K,10K] achieves the highest accuracy in this group ($25.46 \pm 0.71\%$), though given the baseline's standard deviation ($23.28 \pm 1.96\%$), this represents a modest improvement. The two-layer configurations (A2, A4, A6) tend to perform slightly better than single-layer variants, with A4 also showing the best BLEU score ($3.91 \pm 0.17$) and lowest perplexity ($191 \pm 4$) in this group.

**K-layers (Table 5 Rows B):** Temporal encoding layers demonstrate some architectural sensitivity. Configuration B3 [50K,50K] achieves the highest accuracy ($25.52 \pm 0.40\%$) and best BLEU score ($3.93 \pm 0.09$) in this group. While this appears to outperform baseline, the improvement is modest given the measurement variability. The single-layer configuration B1 performs similarly ($25.23 \pm 0.21\%$), while the three-layer setup B4 shows somewhat reduced performance.

**L-layers (Table 5 Rows C):** Shifted target token processing layers show performance generally comparable to baseline, with C2 [16K,8K] and C5 [22K,11K,6K] achieving slightly higher accuracies ($25.12 \pm 0.75\%$ and $25.17 \pm 1.22\%$ respectively). Most configurations in this group maintain performance within the expected variation range of baseline.

**P-layers (Table 5 Rows D):** Autoregressive decoding layers with single large configuration (D3 [96K]) perform best in this group ($25.31 \pm 0.72\%$ accuracy, $3.94 \pm 0.09$ BLEU), showing a modest improvement over baseline. Multi-layer variants (D2, D4, D6) generally perform at or slightly below baseline levels, though differences are within typical measurement variation.

**M-layers (Table 5 Rows E):** Output generation layers show stable performance across configurations, with most variants performing within the noise level of baseline. Configuration E2 [160K,80K,480K] achieves the highest accuracy ($24.87 \pm 0.95\%$) in this group, though the improvement over baseline is marginal.

| | N | K | L | P | M | Emb | k | ACC (%) | BLEU | PPL |
|---|---|---|---|---|---|---|---|---|---|---|
| BASE | 12K,12K | 54K,32K | 12K,12K | 64K,48K | 400K,400K,480K | 1024 | 30 | 23.28±1.96 | 3.59±0.33 | 209±10 |
| A1 | 18K | | | | | | | 24.21±0.59 | 3.76±0.10 | 199±5 |
| A2 | 16K,8K | | | | | | | 24.46±0.96 | 3.76±0.16 | 200±8 |
| A3 | 24K | | | | | | | 24.17±1.13 | 3.75±0.20 | 199±6 |
| A4 | 20K,10K | | | | | | | 25.46±0.71 | 3.91±0.17 | 191±4 |
| A5 | 22K,11K,6K | | | | | | | 24.23±0.58 | 3.75±0.12 | 197±2 |
| A6 | 28K,14K,7K | | | | | | | 25.23±0.59 | 3.87±0.11 | 194±6 |
| B1 | | 48K | | | | | | 25.23±0.21 | 3.89±0.09 | 195±0 |
| B2 | | 50K,28K | | | | | | 24.12±0.73 | 3.80±0.08 | 198±6 |
| B3 | | 50K,50K | | | | | | 25.52±0.40 | 3.93±0.09 | 197±3 |
| B4 | | 50K,48K,38K | | | | | | 24.03±0.71 | 3.70±0.13 | 205±5 |
| B5 | | 80K,60K,48K | | | | | | 24.57±0.80 | 3.80±0.12 | 202±5 |
| C1 | | | 18K | | | | | 23.88±0.45 | 3.66±0.18 | 197±2 |
| C2 | | | 16K,8K | | | | | 25.12±0.75 | 3.85±0.06 | 201±6 |
| C3 | | | 24K | | | | | 23.35±1.08 | 3.59±0.25 | 198±6 |
| C4 | | | 20K,10K | | | | | 24.12±0.72 | 3.73±0.13 | 205±7 |
| C5 | | | 22K,11K,6K | | | | | 25.17±1.22 | 3.89±0.13 | 205±8 |
| C6 | | | 28K,14K,7K | | | | | 24.60±0.26 | 3.82±0.11 | 205±5 |
| D1 | | | | 48K | | | | 24.49±0.22 | 3.80±0.07 | 201±0 |
| D2 | | | | 50K,28K | | | | 23.56±0.39 | 3.60±0.06 | 202±3 |
| D3 | | | | 96K | | | | 25.31±0.72 | 3.94±0.09 | 200±4 |
| D4 | | | | 50K,50K | | | | 24.18±0.35 | 3.69±0.12 | 201±3 |
| D5 | | | | 50K,48K,38K | | | | 24.29±0.66 | 3.74±0.14 | 198±2 |
| D6 | | | | 80K,60K,48K | | | | 23.54±0.96 | 3.55±0.19 | 201±7 |
| E1 | | | | | 120K,60K,480K | | | 24.66±0.89 | 3.82±0.15 | 199±7 |
| E2 | | | | | 160K,80K,480K | | | 24.87±0.95 | 3.89±0.13 | 198±6 |
| E3 | | | | | 240K,480K | | | 24.15±0.48 | 3.75±0.06 | 204±1 |
| E4 | | | | | 200K,100K,50K,480K | | | 24.31±0.18 | 3.71±0.11 | 198±4 |
| E5 | | | | | 480K,480K | | | 23.90±0.55 | 3.64±0.10 | 206±3 |
| E6 | | | | | 320K,160K,80K,480K | | | 24.24±0.76 | 3.71±0.22 | 197±6 |
| F1 | | | | | | 512 | | 24.29±0.62 | 3.69±0.21 | 200±3 |
| F2 | | | | | | 768 | | 24.40±0.87 | 3.77±0.07 | 199±8 |
| F3 | | | | | | 1536 | | 24.82±1.28 | 3.88±0.19 | 199±6 |
| G1 | | | | | 400K,400K,256K | | 16 | 17.55±1.75 | 3.28±0.32 | 293±2 |
| G2 | | | | | 400K,400K,384K | | 24 | 23.62±0.81 | 3.58±0.11 | 216±4 |
| G3 | | | | | 400K,400K,512K | | 32 | 23.80±0.30 | 3.66±0.03 | 197±5 |
| G4 | | | | | 400K,400K,640K | | 40 | 25.06±0.33 | 3.88±0.06 | 188±4 |

Table 5: Impact of layer architecture variations on model test performance. Results show mean ± standard deviation across 3 random seeds. All configurations use the base model hyperparameters unless specified. Architectural components: $N$, $K$, $L$, $P$, $M$, Emb (embedding dimension), $k$ (group factor). Metrics: Accuracy (ACC), BLEU, and perplexity (PPL).

**Embedding Dimension (Table 5 Rows F):** Increasing embedding dimension from 512 to 1536 (F1–F3) shows a gradual trend of improvement. F3 [1536] achieves the highest accuracy ($24.82 \pm 1.28\%$) in this group, though given the large standard deviation, the improvement over baseline is within measurement uncertainty.

**Group Factor (Table 5 Rows G):** Group factor tuning reveals the clearest performance differences in the study. Configuration G1 ($k = 16$) shows substantially lower performance ($17.55 \pm 1.75\%$ accuracy) compared to baseline, while G4 ($k = 40$) achieves better performance ($25.06 \pm 0.33\%$) with lower perplexity ($188 \pm 4$). This represents one of the few cases where the performance differences exceed typical measurement variation.

**Key findings:** (1) Most architectural modifications yield performance changes within measurement uncertainty; (2) Group factor selection shows the clearest impact on performance, with very low values (k=16) notably underperforming; (3) While some configurations (A4, B3, D3) show promising trends, the improvements are generally modest relative to baseline variability; (4) The results suggest that the baseline architecture is reasonably well-tuned, with most alternatives providing only marginal gains.

### C.1.2 INITIALIZATION METHODS

**Node Initialization (Table 6 Rows A):** Node initialization strategies show minor variations around baseline. Gaussian initialization (A1) performs best ($24.51 \pm 0.71\%$ accuracy), while other methods (A2-A4) remain close to baseline levels. Differences are within measurement uncertainty.

| | Node Residual | Hidden State Gaussian | ACC | BLEU | PPL |
|---|---|---|---|---|---|
| Base | Residual | Gaussian | 23.28±1.96 | 3.59±0.33 | 209±10 |
| A1 | | Gaussian | 24.51±0.71 | 3.83±0.08 | 199±6 |
| A2 | | One | 24.26±1.31 | 3.65±0.24 | 203±11 |
| A3 | | Uniform | 23.72±0.40 | 3.61±0.11 | 208±2 |
| A4 | | Zero | 24.30±1.47 | 3.77±0.15 | 201±6 |
| B1 | Gaussian | | 22.59±0.89 | 3.35±0.14 | 260±10 |
| B2 | Residual | | 24.63±0.42 | 3.79±0.16 | 199±4 |

Table 6: Impact of initialization methods on model test performance. Comparison of different node initialization and hidden state initialization strategies. Metrics show validation mean ± standard deviation across multiple runs with 3 different seeds.

**Hidden State Initialization (Table 6 Rows B):** Hidden state initialization shows clearer effects. Gaussian initialization (B1) notably degrades performance (22.59 ± 0.89% accuracy, 260 ± 10 perplexity), while Residual initialization (B2) maintains baseline performance (24.63 ± 0.42%).

**Key findings:** Node initialization methods yield minimal differences, while hidden state initialization significantly impacts performance, with Gaussian initialization being detrimental compared to the baseline Residual approach.

## C.2 Training Parameter

### C.2.1 Learning Rate and Gradients

| | LR | WD | N | K | L | P | M | ACC (%) | BLEU | PPL |
|---|---|---|---|---|---|---|---|---|---|---|
| Base | 0.050 | 0.001 | | | | | | 23.28±1.96 | 3.59±0.33 | 209±10 |
| A1 | 0.001 | | | | | | | 0.44±0.03 | 0.21±0.08 | 12729±418 |
| A2 | 0.010 | | | | | | | 12.36±0.21 | 1.74±0.03 | 987±36 |
| A3 | 0.050 | | | | | | | 23.66±1.96 | 3.65±0.33 | 203±12 |
| A4 | 0.100 | | | | | | | 24.42±0.76 | 3.83±0.14 | 205±5 |
| A5 | 0.500 | | | | | | | 22.46±0.42 | 3.47±0.10 | 266±7 |
| B1 | | 1.0E-05 | | | | | | 24.04±0.72 | 3.68±0.15 | 205±8 |
| B2 | | 1.0E-04 | | | | | | 24.06±0.84 | 3.63±0.16 | 205±7 |
| B3 | | 5.0E-04 | | | | | | 23.67±0.47 | 3.62±0.15 | 207±4 |
| B4 | | 0.001 | | | | | | 23.49±0.82 | 3.67±0.13 | 203±5 |
| B5 | | 0.010 | | | | | | 23.59±0.86 | 3.63±0.17 | 205±7 |
| C1 | | | | | | | | 23.79±1.81 | 3.66±0.30 | 203±10 |
| C2 | | | ✽ | | | | | 22.69±0.53 | 3.52±0.15 | 208±6 |
| C3 | | | | ✽ | | | | 23.37±1.00 | 3.60±0.23 | 208±8 |
| C4 | | | | | ✽ | | | 22.64±0.71 | 3.53±0.15 | 214±4 |
| C5 | | | | | | ✽ | | 21.80±0.26 | 3.28±0.06 | 236±2 |
| C6 | | | | | | | ✽ | 23.58±0.16 | 3.61±0.05 | 223±1 |

Table 7: Impact of learning rate, weight decay, and frozen layers on model performance. LR denotes learning rate, WD denotes weight decay. ✽ indicates frozen layer groups. Metrics are test mean ± std across runs with different seeds.

**Learning Rate (Table 7 Rows A):** Learning rate selection shows strong sensitivity. Very low rates (A1: 0.001) completely fail to train (0.44 ± 0.03% accuracy), while moderate rates (A2: 0.010) severely underperform (12.36 ± 0.21%). The baseline rate (A3: 0.050) and higher rate (A4: 0.100) perform similarly well, with A4 showing modest improvement (24.42 ± 0.76%). Very high rates (A5: 0.500) degrade performance slightly.

**Weight Decay (Table 7 Rows B):** Weight decay variations show minimal impact on performance. All tested values (B1-B5) yield accuracies within $\pm 1\%$ of baseline, with differences well within measurement uncertainty. The baseline value (0.001) appears adequate.

**Frozen Layers (Table 7 Rows C):** Freezing individual layer groups generally maintains or slightly reduces performance. Freezing P-layers (C5) shows the largest degradation ($21.80 \pm 0.26\%$ accuracy), while other frozen configurations (C2-C4, C6) remain close to baseline levels. All frozen configurations underperform the fully trainable baseline (C1).

**Key findings:** Learning rate is critical, with very low values causing training failure and moderate-to-high values (0.050-0.100) working well. Weight decay has minimal impact within reasonable ranges. Freezing any layer group slightly degrades performance, suggesting all components benefit from training.

## C.3 STOCHASTIC COMPONENTS

| | DROPOUT | LABEL SMOOTH | GUMBEL $\tau$ | GROUP SUM $\tau$ | ACC (%) | BLEU | PPL |
|---|---|---|---|---|---|---|---|
| BASE | 0 | 0.1 | | 2 | 23.28±1.96 | 3.59±0.33 | 209±10 |
| A2 | 0.05 | | | | 23.33±1.36 | 3.52±0.31 | 231±11 |
| A3 | 0.1 | | | | 22.07±1.09 | 3.41±0.24 | 277±20 |
| A4 | 0.2 | | | | 21.32±0.80 | 3.41±0.18 | 378±26 |
| A5 | 0.3 | | | | 14.54±7.06 | 2.44±0.99 | 1146±366 |
| B1 | | 0 | | | 23.03±0.46 | 3.55±0.11 | 211±1 |
| B2 | | 0.01 | | | 23.93±0.18 | 3.74±0.11 | 208±1 |
| B4 | | 0.2 | | | 23.91±0.44 | 3.58±0.10 | 210±4 |
| B5 | | 0.5 | | | 23.38±0.57 | 3.60±0.04 | 267±5 |
| C1 | | | 0.01 | | 19.76±1.21 | 2.96±0.24 | 533±20 |
| C2 | | | 0.1 | | 20.78±0.78 | 3.13±0.09 | 362±12 |
| C3 | | | 0.5 | | 20.42±0.28 | 3.06±0.04 | 307±8 |
| C4 | | | 1 | | 19.82±0.20 | 2.95±0.02 | 309±5 |
| D1 | | | | 0.25 | 4.45±0.86 | 1.04±0.16 | 46932±16532 |
| D2 | | | | 0.5 | 16.44±1.25 | 2.90±0.20 | 769±83 |
| D3 | | | | 1 | 22.66±0.88 | 3.61±0.17 | 229±2 |
| D5 | | | | 4 | 21.01±0.13 | 2.58±0.01 | 323±1 |
| D6 | | | | 8 | 20.93±0.00 | 2.58±0.00 | 1444±3 |

Table 8: Impact of regularization and loss parameters on model performance. Metrics are mean ± std on validation set.

**Dropout (Table 8 Rows A):** Dropout regularization is applied after each logic layer group (N, K, L, P, M layers) and after embedding, with separate dropout modules for each layer group. For a given layer output $\mathbf{h}$, dropout applies:

$$\mathbf{h}_{\text{dropout}} = \mathbf{h} \odot \mathbf{m}$$

where $\mathbf{m} \sim \text{Bernoulli}(1-p)$ is a binary mask and $p$ is the dropout probability. Light dropout (A2: 0.05) maintains baseline performance, while moderate dropout (A3: 0.1, A4: 0.2) progressively degrades accuracy and increases perplexity. Heavy dropout (A5: 0.3) severely impairs training ($14.54 \pm 7.06\%$ accuracy), suggesting the model's logic layers are particularly sensitive to regularization, likely because dropout disrupts the discrete logic operations.

**Label Smoothing (Table 8 Rows B):** Label smoothing modifies the target distribution by mixing the one-hot encoded labels with a uniform distribution:

$$\mathbf{y}_{\text{smooth}} = (1 - \alpha)\mathbf{y}_{\text{true}} + \frac{\alpha}{K}\mathbf{1}$$

where $\alpha$ is the smoothing parameter, $K$ is the number of classes, and $\mathbf{1}$ is the all-ones vector. Light smoothing (B2: 0.01) achieves slight improvement ($23.93 \pm 0.18\%$), while other values (B1, B4, B5) remain close to baseline. The baseline value (0.1) appears appropriate.

**Gumbel Temperature (Table 8 Rows C):** The Gumbel-Softmax temperature parameter controls the stochasticity of logic gate selection in the neural logic layers. Each neuron computes $\phi_j = f_{\mathbf{z}_j}(x_{i_j}, x_{k_j})$ using softmax weights. With Gumbel-Softmax, this becomes:

$$\phi_j = \sum_{i=0}^{15} \frac{\exp((z_{j,i} + G_i)/\tau)}{\sum_{\ell=0}^{15} \exp((z_{j,\ell} + G_\ell)/\tau)} \cdot g_i(x_{i_j}, x_{k_j})$$

where $G_i \sim -\log(-\log(\mathcal{U}(0,1)))$ is Gumbel noise and $\tau$ is the temperature parameter. Lower temperatures make selections more discrete, while higher temperatures increase randomness. All tested values (C1-C4) substantially underperform baseline, with C2 ($\tau = 0.1$) performing best in this group ($20.78 \pm 0.78\%$) but still notably below baseline performance.

**Group Sum Temperature (Table 8 Rows D):** Group sum temperature modifies the GroupSum operation. The temperature-scaled GroupSum becomes:

$$\text{GroupSum}_\tau(\mathbf{x}) = \frac{1}{\tau} \sum_{j=0}^{d/k-1} \mathbf{x}_{j \cdot k:(j+1) \cdot k}$$

where $\tau$ controls the magnitude of the aggregated outputs before softmax normalization. Very low values (D1: 0.25) cause training failure ($4.45 \pm 0.86\%$) by making the pre-softmax logits too large, leading to numerical instability. Moderate values around the baseline (D3: $\tau = 1$) maintain good performance ($22.66 \pm 0.88\%$). Higher temperatures (D5, D6) progressively degrade performance by making the logits too small, reducing the model's ability to make confident predictions.

**Key findings:** The model shows high sensitivity to stochastic components. Dropout, applied after each layer group, should be minimal or avoided due to its interference with discrete logic operations. Label smoothing has little impact within reasonable ranges. Gumbel-Softmax consistently underperforms the baseline discrete approach, suggesting deterministic gate selection works better for logic-based architectures. Group sum temperature is critical for numerical stability and prediction confidence, with values around 1-2 being optimal.

# D   COMPUTATIONAL SETUP

The model was trained for 130K steps on NVIDIA GeForce RTX 3090 and NVIDIA TITAN Xp GPUs, with average step times of 0.2 seconds and 0.6 seconds, respectively, corresponding to total training times of approximately 7.2 hours and 21.7 hours.

