# OpenReview forum: "Recurrent Deep Differentiable Logic Gate Networks"
_ICLR.cc/2026/Conference — Submitted to ICLR 2026_

### Official Review · Reviewer_3duw · 2025-10-30

**Soundness:** 1
**Presentation:** 2
**Contribution:** 1
**Rating:** 0
**Confidence:** 5

**Summary:**

Deep Differentiable Logic Gate Networks (DDLGNs) are extended by recurrency, and then an encoder-decoder model is build based on that.

The model is tested on the WMT 2014 English-to-German translation task.

**Strengths:**

* Interesting novel architecture.
* Good motivation.
* Shifted monolingual prediction result is interesting.
* Some good ablations in the appendix.

**Weaknesses:**

* The experiments seems flawed. The paper reports about 5 BLEU on WMT’14 English-German translation. Usual numbers on this task are about 30-35 BLEU. So this is completely broken?
* Sequence lengths are way too short for reasonable realistic experiments.
* Models are way too small.
* Contradiction on vanishing gradients (see comments below).
* Contradiction on long-sequence handling (see comments below).
* Unclear parts.

**Questions:**

I don't understand the BLEU scores. Is that serious? Is that correct? Do you maybe measure sth different? They are far away from usual number that you would expect, which are in the range of 30-35 BLEU. They are so far off that this is either measured incorrectly, or measured sth differently, or totally broken. No matter what it is, it basically makes the whole work meaningless.

Why no cross attention, as it would be common for encoder-decoder models?

"The data is tokenized at the word level using regex-based splitting, with a shared
16,000-token vocabulary for English and German." - I don't understand what this means. Do you use BPE or SPM or sth like that?

The models are obviously way too small.

If the goal is an efficient model, you should show how it performs when you give it the same amount of compute as a normal-sized well-performing baseline (e.g. some reasonable-sized Transformer). Does it perform better then?

"sequence length of 16 tokens" - this just makes it a toy task. Increase it to at least 50-100 or so to make it actually interesting and relevant.

Contradiction on vanishing gradients:
Sec 5.5: "confirming the absence of vanishing or exploding gradients"
Sec 6: "Additionally, the architecture suffers from vanishing gradient problems"

Contradiction on long-sequence handling:
The shifted-token task is used to argue that RDDLGNs have superior long-range memorization capabilities (Fig 4).
The tokenizer ablation shows the model's performance collapses as sequence length increases (Table 2).

**Details Of Ethics Concerns:**

.

---

> ### Author Response · Authors · 2025-11-13
>
> We thank the reviewer for their time and attention. The reported bleu scores were calculated using the calculate_bleu function in src/metrics/machine_translation.py lines 20 to 33. This used `NLTK` to compute smoothed sentence-level scores that were averaged to give the final numbers. However, there was a bug in how the tokens were handled before feeding the predictions and targets to the `NLTK`.
> We reran the experiments and logged the sentence and corpus-level BLEU scores calculated using the `sacrebleu` library’s functions `sacrebleu.corpus_bleu` and `sacrebleu.sentence_bleu`, respectively.
>
> Here are the updated values:
>
> |           Model         | sentence_bleu | corpus_bleu |
> |-------------------------|-------------|---------------|
> | GRU                     |    19.59    |     20.70     |
> | RNN                     |    18.38    |     19.15     |
> | LSTM                    |    19.79    |     20.74     |
> | Transformer             |    18.41    |     19.27     |
> | RDDLGN                  |    18.20    |     18.90     |
> | RDDLGN (collapsed)      |    16.90    |     17.68     |
>
>
> We have updated the supplementary material with the updated configs that enables logging of sacrebleu, and we will update the pdf with these numbers.
>
> We will also provide numbers for longer contexts but wanted to provide the above numbers as soon as possible. While 16 tokens is more in the realm of toy tasks, DLGNs are still in their infancy. Similar comments could be said of the paper by Petersen et al. 2022, where the CIFAR10 accuracy is 50-60% with their follow up work in 2024 bringing it to 80%.
>
> Regarding the choice of baselines. Here we followed the practice of Petersen et al. 2022. Differentiable logic gate networks (DLGNs) are still hard to scale to large model sizes, so we scaled the baseline to be comparable in model size. However, note that the bleu scores ~5 were not caused by the model sizes but rather the way bleu was measured.
>
> We use word-level tokenization via regex-based splitting (\w+|\S pattern), building a frequency-based vocabulary of the 16,000 most common tokens. Unknown words are mapped to <UNK>. This differs from subword methods (BPE/SPM) which are available as alternative tokenization levels in our implementation.
>
> Regarding the vanishing gradients, we will rephrase Sec. 5.5 and 6 to make the text clearer. Sec. 5.5 shows that our models at their current size are not suffering from vanishing gradients. However, DLGNs are difficult to scale as the addition of extra layers can cause sudden and significant gradient issues.
>
> Lastly, regarding the comment on the long-sequence handling. The experiments test the model at different scales and for different capabilities. We will run the shift factor experiment with longer factors.

---

> ### Author Response · Authors · 2025-11-21
>
> After discussions with reviewer U2Mo, we found that the cause of the low BLEU scores was a bug in how the predictions and targets were given to the relevant `NLTK` function. We have updated our previous comment with this in mind. The BLEU scores are:
>
> |           Model         | sentence_bleu | corpus_bleu |
> |-------------------------|-------------|---------------|
> | GRU                     |    19.59    |     20.70     |
> | RNN                     |    18.38    |     19.15     |
> | LSTM                    |    19.79    |     20.74     |
> | Transformer             |    18.41    |     19.27     |
> | RDDLGN                  |    18.20    |     18.90     |
> | RDDLGN (collapsed)      |    16.90    |     17.68     |
>
> We will rerun the ablations and then update the PDF with the new BLEU scores.
>
> We tested larger shift factors (up to 48) and the result of this experiment can be seen below. All models exhibit a substantial decline in memorization performance under larger shifts. The RDDLGN maintains strong performance only for small shifts, which is consistent with its fixed hidden vector size (32'000) that limits the amount of information it can propagate across temporal steps.
>
> | Model  | 0        | 8        | 16       | 32       | 48       |
> | ------ | -------- | -------- | -------- | -------- | -------- |
> | GRU    | 0.9772   | 0.2247   | 0.1237   | 0.0550   | 0.0592   |
> | RNN    | 0.9858   | 0.0957   | 0.0605   | 0.0609   | 0.0588   |
> | RDDLGN | 0.9781   | 0.4526   | 0.0788   | 0.0622   | 0.0601   |
>
>
>
>
> Please let us know if there are any further comments or inputs; we are happy to continue the discussion. If you are content with the improvements of our submission, we would appreciate it if you could reflect this in your score.

---

### Official Review · Reviewer_dcov · 2025-11-03

**Soundness:** 2
**Presentation:** 2
**Contribution:** 2
**Rating:** 0
**Confidence:** 3

**Summary:**

This paper introduces Recurrent Deep Differentiable Logic Gate Networks (RDDLGN)—a recurrent seq2seq model whose layers are mixtures of relaxed Boolean operators and which can be “collapsed” after training into a purely logical (binary) network intended for efficient inference (e.g., on FPGAs). On WMT’14 En→De, the uncollapsed model reports 5.00 BLEU / 30.9% token accuracy, and the collapsed variant reports 4.39 BLEU / 27.7%, roughly between vanilla RNN and GRU baselines under the authors’ settings.

**Strengths:**

The paper extends DDLGNs to sequence modeling with an encoder–decoder built from logic layers.

The memory evaluation is interesting. The shifted-copy task shows RDDLGN maintains high accuracy at larger temporal offsets than RNN/GRU,

**Weaknesses:**

Baselines are quite small and non-standard for WMT14, 5-ish BLEU feel like garbage and not convincing it's actually a practical setup.

**Questions:**

I'd suggest the author why using such MT setup.

---

> ### Author Response · Authors · 2025-11-13
>
> We thank the reviewer for their time and attention. The reported bleu scores were calculated using the calculate_bleu function in src/metrics/machine_translation.py lines 20 to 33. This used `NLTK` to compute smoothed sentence-level scores that were averaged to give the final numbers. However, there was a bug in how the tokens were handled before feeding the predictions and targets to the `NLTK`.
> We reran the experiments and logged the sentence and corpus-level BLEU scores calculated using the `sacrebleu` library’s functions `sacrebleu.corpus_bleu` and `sacrebleu.sentence_bleu`, respectively.
>
> Here are the updated values:
>
> |           Model         | sentence_bleu | corpus_bleu |
> |-------------------------|-------------|---------------|
> | GRU                     |    19.59    |     20.70     |
> | RNN                     |    18.38    |     19.15     |
> | LSTM                    |    19.79    |     20.74     |
> | Transformer             |    18.41    |     19.27     |
> | RDDLGN                  |    18.20    |     18.90     |
> | RDDLGN (collapsed)      |    16.90    |     17.68     |
>
>
> We have updated the supplementary material with the updated configs that enables logging of sacrebleu, and we will update the pdf with these numbers.
>
>
> Regarding the choice of baselines. Here we followed the practice of Petersen et al. 2022. Differentiable logic gate networks are still hard to scale to large model sizes, so we scaled the baseline to be comparable in model size. Are there any other specific baselines we should include?

---

> ### Author Response · Authors · 2025-11-21
>
> After discussions with reviewer U2Mo, we found that the cause of the low BLEU scores was a bug in how the predictions and targets were given to the relevant `NLTK` function. We have updated our previous comment with this in mind. The BLEU scores are:
>
> |           Model         | sentence_bleu | corpus_bleu |
> |-------------------------|-------------|---------------|
> | GRU                     |    19.59    |     20.70     |
> | RNN                     |    18.38    |     19.15     |
> | LSTM                    |    19.79    |     20.74     |
> | Transformer             |    18.41    |     19.27     |
> | RDDLGN                  |    18.20    |     18.90     |
> | RDDLGN (collapsed)      |    16.90    |     17.68     |
>
> We will rerun the ablations and then update the PDF with the new BLEU scores.
>
> Please let us know if there are any further comments or inputs; we are happy to continue the discussion. If you are content with the improvements of our submission, we would appreciate it if you could reflect this in your score.

---

### Official Review · Reviewer_U2Mo · 2025-11-10

**Soundness:** 1
**Presentation:** 2
**Contribution:** 2
**Rating:** 0
**Confidence:** 3

**Summary:**

This review is written by the AC after reading the 2 reviews with 0 scores. Since there is one major, likely fatal issue raised by both the reviewers, it is appropriate to focus on that instead of an independent emergency reviewer. The issue is that the bleu score reported in this paper are around 5 (as opposed to 20-40), if the reviewer's interpretation is correct, this would makes any comparisons in the paper meaningless irrespective of what method was proposed, or what the exact setting was. The authors did highlight these bleu scores in the abstract, showing this result is key to the paper as opposed to a side experiment.

To provide some specific references, On the EN-DE newstest task, phrased based system from 2014 had ~20 bleu score:
https://web.archive.org/web/20140625052707/http://matrix.statmt.org/
 transformer reported 28 in 2017.
Table 25-26 of the "Findings of WMT 2014" https://aclanthology.org/W14-3302.pdf for the medical tasks also reported ~20 bleu scores.

~5 bleu score was reported in tables 2 and 3 of this submission, which is probably around a word dictionary look up.

The authors should respond to this in the response period.

**Strengths:**

not evaluated

**Weaknesses:**

not evaluated

**Questions:**

Can the authors explain why their range of BLEU scores are out of normal, see summary for specific pointers?

---

> ### Author Response · Authors · 2025-11-13
> **The low bleu scores in the papers**
>
> We thank the reviewers for their time and attention. The reported bleu scores were calculated using the `calculate_bleu` function in `src/metrics/machine_translation.py` lines 20 to 33. This used `NLTK` to compute smoothened sentence level scores that were averaged to give the final numbers. Therefore, the reported numbers are not directly comparable to bleu scores in other papers.
>
> We reran the experiments and logged the bleu score calculated using the `sacrebleu` library (see lines 35 to 49 in the same file). With this we get the following values:
>
> | GRU | Transformer | RNN | LSTM |  RDDLGN (Collapsed) |
> |-----|-------------|-----|------|---------------------|
> | 20.8|     18.9    | 18.7| 20.4 |        19.2         |
>
> We have updated the supplementary material with the updated configs that enables logging of sacrebleu, and we will update the pdf with these numbers.
>
> We will also provide numbers for longer contexts but wanted to provide the above numbers as soon as possible.
>
> We hope this addresses the reviewers' concerns regarding the bleu scores.

---

> > ### Comment · Reviewer_U2Mo · 2025-11-17
> >
> > Slightly not comparable is possible, but going from 5 to 20 probably requires an explanation based on the settings used and should not be due to subtle difference of NLTK vs scarebleu.

---

> ### Author Response · Authors · 2025-11-18
>
> The difference comes from the change in what is measured. With nltk, we measure sentence-level BLEU scores, while with sacrebleu, we measure corpus-level BLEU scores. The former method has the problem that the numbers are not comparable to those in other papers, while the goal of sacrebleu is to standardize the evaluation, making the numbers comparable with other papers [1].
>
> Regarding the size of the change: If we look at the work by Chen et al. (2014) [Table 3, 2], where they compare smoothing functions, then we see that ground truth sentence level is around 12 to 19, with corpus level around 71 to 73 (depending on smoothing functions). Thus, the relative differences we see are entirely expected from these numbers.
>
> [1] https://arxiv.org/pdf/1804.08771
>
> [2] Chen, Boxing, and Colin Cherry. "A systematic comparison of smoothing techniques for sentence-level BLEU." Proceedings of the ninth workshop on statistical machine translation. 2014.

---

> ### Comment · Reviewer_U2Mo · 2025-11-20
>
> The claims in the rebuttal is misleading. Table 3 and 2 of [2] measures the correlation of the BLEU scores with human judgements, not sentence vs corpus level bleu. [1] shows the various implementation, while already concerning, usually only leads to a few BLEU points of difference, mostly < 1 point except for UNK.
>
> [1] https://arxiv.org/pdf/1804.08771
>
> [2] Chen, Boxing, and Colin Cherry. "A systematic comparison of smoothing techniques for sentence-level BLEU." Proceedings of the ninth workshop on statistical machine translation. 2014.

---

> ### Author Response · Authors · 2025-11-21
>
> Thank you for pointing this out! We had misunderstood the tables.
> We went over the code and experiments again and redid the BLEU calculations. There was a bug in how the code converted tokens to plain text before calculating the BLEU score. After fixing this and using `sacrebleu` to calculate sentence-level **and** corpus-level BLEU scores, the scores are now aligned.
>
> To make everything easier for the reviewers to find and for complete transparency, we provide the new function used to calculate BLEU scores below. The old functions remain in the codebase. While our understanding of what the expected difference between sentence and corpus-level metrics, and Tables 2 and 3 in [2]  were flawed, the corpus-level values provided here in the rebuttal were correct.
>
> Here are the updated values:
> |           Model         | sentence_bleu | corpus_bleu |
> |-------------------------|-------------|---------------|
> | GRU                     |    19.59    |     20.70     |
> | RNN                     |    18.38    |     19.15     |
> | LSTM                    |    19.79    |     20.74     |
> | Transformer             |    18.41    |     19.27     |
> | RDDLGN                  |    18.20    |     18.90     |
> | RDDLGN (collapsed)      |    16.90    |     17.68     |
>
> We will rerun the ablations to get the updated BLEU scores and then update the PDF.
>
> Please let us know if there are any further comments or inputs; we are happy to continue the discussion.
>
> ```
> def calculate_bleu_scores_corrected(logits, targets, vocab, tokenizer, pad_index=0):
>     """
>     Calculates BLEU on DETOKENIZED (human-readable) text.
>
>     Args:
>         tokenizer: Optional tokenizer object with decode() method for proper detokenization
>     """
>     idx2token = _build_idx2token(vocab)
>     pred_ids = logits.argmax(-1).cpu().numpy()
>     targ_ids = targets.cpu().numpy()
>
>     pred_strs = []
>     targ_strs = []
>
>     for i, (pred, targ) in enumerate(zip(pred_ids, targ_ids)):
>         # Get non-padded token IDs
>         pred_ids_clean = _remove_padding(pred, pad_index)
>         targ_ids_clean = _remove_padding(targ, pad_index)
>
>         # Detokenize using tokenizer
>         pred_str = tokenizer.decode(pred_ids_clean)
>         targ_str = tokenizer.decode(targ_ids_clean)
>
>         pred_strs.append(pred_str)
>         targ_strs.append(targ_str)
>
>     # 3. Calculate Scores
>     # SacreBLEU handles its own internal tokenization, so we feed it raw text.
>     corpus_score = sacrebleu.corpus_bleu(pred_strs, [targ_strs]).score
>
>     # Calculate Average Sentence BLEU properly
>     sent_scores = [sacrebleu.sentence_bleu(p, [t]).score for p, t in zip(pred_strs, targ_strs)]
>     avg_sent_score = sum(sent_scores) / len(sent_scores) if sent_scores else 0.0
>
>     return corpus_score, avg_sent_score
> ```

---

### Author Response · Authors · 2025-12-03

Given the changes to the rebuttal, we would like to provide a summary of the discussion.

The two reviewers and the AC highlighted one main weakness: the BLEU score and the choice of baselines. One reviewer also requested that the memorization experiment be done for longer shifts.

For the original submission, there was a bug in how the text was tokenized when calculating the BLEU score using the `nltk` library that shifted the values down. We fixed this using the `sacrebleu` library, and the updated values can be found below. We still need to run all the ablations again, and will update the paper with the new values once this has been done. However, this did not affect any of the conclusions drawn in the paper.

|           Model         | sentence_bleu | corpus_bleu |
|-------------------------|-------------|---------------|
| GRU                     |    19.59    |     20.70     |
| RNN                     |    18.38    |     19.15     |
| LSTM                    |    19.79    |     20.74     |
| Transformer             |    18.41    |     19.27     |
| RDDLGN                  |    18.20    |     18.90     |
| RDDLGN (collapsed)      |    16.90    |     17.68     |

We also tested larger shift factors (up to 48), and the result of this experiment can be seen below. The table shows the test accuracies similar to Figure 4. All models exhibit a substantial decline in memorization performance under larger shifts. The RDDLGN maintains strong performance only for small shifts, which is consistent with its fixed hidden vector size that limits the amount of information it can propagate across temporal steps.

| Model  | 0        | 8        | 16       | 32       | 48       |
| ------ | -------- | -------- | -------- | -------- | -------- |
| GRU    | 0.9772   | 0.2247   | 0.1237   | 0.0550   | 0.0592   |
| RNN    | 0.9858   | 0.0957   | 0.0605   | 0.0609   | 0.0588   |
| RDDLGN | 0.9781   | 0.4526   | 0.0788   | 0.0622   | 0.0601   |

Regarding the choice of baselines. Here, we followed the practice of Petersen et al. 2022. Differentiable logic gate networks (DLGNs) are still hard to scale to large model sizes, so we scaled the baseline to be comparable in model size.

While 16 tokens are more in the realm of toy tasks, DLGNs are still in their infancy. Similar comments could be said of the paper by Petersen et al. 2022, where the CIFAR10 accuracy is 50-60% with their follow-up work in 2024, bringing it to 80%.

DLGNs are currently focused on working well in low-resource settings. However, they also allow for ultra-fast inference, so they can also work well in domains where low latency is essential.

At the moment, DLGNs achieve lower accuracy compared to modern CNNs, Transformers, etc., but these models have benefited from over two decades of intensive research, engineering, and community-wide innovation. In contrast, DLGNs are a very new class of models, still in their infancy, so, unsurprisingly, there is currently a performance gap. We are not claiming that DLGNs will necessarily reach the level of CNNs. However, we expect that gap to shrink as the field of differentiable logic networks matures.

---

### Meta-Review · Area_Chair_ax1W · 2026-01-04

**Summary:**

This paper introduces Recurrent Deep Differentiable Logic Gate Networks (RDDLGN), a novel architecture that combines differentiable Boolean operations with recurrent neural networks for sequence-to-sequence learning. The authors apply this method to the WMT'14 English-German translation task, aiming to establish the viability of recurrent logic-based neural computation for potential hardware acceleration.

**Reviewer Concerns:**

**1. Critical Flaws in Experimental Results (BLEU Scores)**
The primary failure mode of this submission was the reporting of translation results that were drastically below standard baselines, rendering the evaluation meaningless.

* **Implausible Results:** The submitted paper reported a BLEU score of 5.00. Reviewers U2Mo and 3duw pointed out that standard performance for this task is typically in the range of 20 to 30+ BLEU.

* **Lack of Sanity Checking:** Reviewers noted that a score of 5 (versus expected baselines) should have triggered an immediate internal investigation by the authors before submission. As Reviewer U2Mo noted, a score of 5 is comparable to simple word dictionary lookups.

**2. Inadequate Experimental Setup**
Beyond the metric calculation error, reviewers raised significant concerns regarding the rigor of the setup:

* **Toy-Task Scaling:** Reviewer 3duw and Reviewer dcov noted that the sequence length was capped at 16 tokens, which effectively reduces a complex translation benchmark to a "toy task".


* **Baselines:** The baselines were criticized for being too small and non-standard to provide a convincing comparison for practical utility.


**3. Internal Contradictions**
Reviewer 3duw identified contradictions in the text, specifically regarding vanishing gradients. The paper claims an absence of vanishing gradients in Section 5.5, yet Section 6 admits the architecture suffers from them.

**Reviewer Scores:**

0

---

### Decision · Program_Chairs · 2026-01-26

Reject